# Sketching Structured Matrices for Faster Nonlinear Regression

**Haim Avron    Vikas Sindhwani**
IBM T.J. Watson Research Center
Yorktown Heights, NY 10598
{haimav,vsindhw}@us.ibm.com

**David P. Woodruff**
IBM Almaden Research Center
San Jose, CA 95120
dpwoodru@us.ibm.com

## Abstract

Motivated by the desire to extend fast randomized techniques to nonlinear $l_p$ regression, we consider a class of structured regression problems. These problems involve Vandermonde matrices which arise naturally in various statistical modeling settings, including classical polynomial fitting problems, additive models and approximations to recently developed randomized techniques for scalable kernel methods. We show that this structure can be exploited to further accelerate the solution of the regression problem, achieving running times that are faster than "input sparsity". We present empirical results confirming both the practical value of our modeling framework, as well as speedup benefits of randomized regression.

## 1   Introduction

Recent literature has advocated the use of randomization as a key algorithmic device with which to dramatically accelerate statistical learning with $l_p$ regression or low-rank matrix approximation techniques [12, 6, 8, 10]. Consider the following class of regression problems,

$$\arg\min_{x \in \mathcal{C}} \|Zx - b\|_p, \text{ where } p = 1, 2 \tag{1}$$

where $\mathcal{C}$ is a convex constraint set, $Z \in \mathbb{R}^{n \times k}$ is a sample-by-feature design matrix, and $b \in \mathbb{R}^n$ is the target vector. We assume henceforth that the number of samples is large relative to data dimensionality ($n \gg k$). The setting $p = 2$ corresponds to classical least squares regression, while $p = 1$ leads to least absolute deviations fit, which is of significant interest due to its robustness properties. The constraint set $\mathcal{C}$ can incorporate regularization. When $\mathcal{C} = \mathbb{R}^k$ and $p = 2$, an $\epsilon$-optimal solution can be obtained in time $O(nk \log k) + \text{poly}(k \, \epsilon^{-1})$ using randomization [6, 19], which is much faster than an $O(nk^2)$ deterministic solver when $\epsilon$ is not too small (dependence on $\epsilon$ can be improved to $O(\log(1/\epsilon))$ if higher accuracy is needed [17]). Similarly, a randomized solver for $l_1$ regression runs in time $O(nk \log n) + \text{poly}(k \, \epsilon^{-1})$ [5].

In many settings, what makes such acceleration possible is the existence of a suitable *oblivious subspace embedding* (OSE). An OSE can be thought of as a data-independent random "sketching" matrix $S \in \mathbb{R}^{t \times n}$ whose approximate isometry properties over a subspace (e.g., over the column space of $Z, b$) imply that,

$$\|S(Zx - b)\|_p \approx \|Zx - b\|_p \text{ for all } x \in \mathcal{C} ,$$

which in turn allows $x$ to be optimized over a "sketched" dataset of much smaller size without losing solution quality. Sketching matrices include Gaussian random matrices, structured random matrices which admit fast matrix multiplication via FFT-like operations, and others.

This paper is motivated by two questions which in our context turn out to be complimentary:

○ Can *additional structure* in $Z$ be non-trivially exploited to further accelerate runtime? Clarkson and Woodruff have recently shown that when $Z$ is sparse and has $\texttt{nnz}(Z) \ll nk$ non-zeros, it is possible to achieve much faster "input-sparsity" runtime using hashing-based sketching matrices [7]. Is it possible to further beat this time in the presence of additional structure on $Z$?

○ Can faster and more accurate sketching techniques be designed for *nonlinear and nonparametric regression*? To see that this is intertwined with the previous question, consider the basic problem of fitting a polynomial model, $b = \sum_{i=1}^{q} \beta_i z^i$ to a set of samples $(z_i, b_i) \in \mathbb{R} \times \mathbb{R}, i = 1, \ldots, n$. Then, the design matrix $Z$ has Vandermonde structure which can potentially be exploited in a regression solver. It is particularly appealing to estimate non-parametric models on large datasets. Sketching algorithms have recently been explored in the context of kernel methods for non-parametric function estimation [16, 11].

To be able to precisely describe the structure on $Z$ that we consider in this paper, and outline our contributions, we need the following definitions.

**Definition 1** *(Vandermonde Matrix) Let $x_0, x_1, \ldots, x_{n-1}$ be real numbers. The Vandermonde matrix, denoted $V_{q,n}(x_0, x_1, \ldots, x_{n-1})$, has the form:*

$$V_{q,n}(x_1, x_1, \ldots, x_{n-1}) = \begin{pmatrix} 1 & 1 & \ldots & 1 \\ x_0 & x_1 & \ldots & x_{n-1} \\ \ldots & \ldots & \ldots & \ldots \\ x_0^{q-1} & x_1^{q-1} & \ldots & x_{n-1}^{q-1} \end{pmatrix}$$

Vandermonde matrices of dimension $q \times n$ require only $O(n)$ implicit storage and admit $O((n + q) \log^2 q)$ matrix-vector multiplication time. We also define the following matrix operator $T_q$ which maps a matrix $A$ to a *block-Vandermonde* structured matrix.

**Definition 2** *(Matrix Operator) Given a matrix $A \in \mathbb{R}^{n \times d}$, we define the following matrix:*

$$T_q(A) = \left[ \ V_{q,n}(A_{1,1}, \ldots, A_{n,1})^T \mid V_{q,n}(A_{1,2}, \ldots, A_{n,2})^T \mid \cdots \mid V_{q,n}(A_{1,d}, \ldots, A_{n,d})^T \ \right]$$

In this paper, we consider regression problems, Eqn. 1, where $Z$ can be written as

$$Z = T_q(A) \tag{2}$$

for an $n \times d$ matrix $A$, so that $k = dq$. The operator $T_q$ expands each feature (column) of the original dataset $A$ to $q$ columns of $Z$ by applying monomial transformations upto degree $q - 1$. This lends a block-Vandermonde structure to $Z$. Such structure naturally arises in polynomial regression problems, but also applies more broadly to non-parametric additive models and kernel methods as we discuss below. With this setup, the goal is to solve the following problem:

**Structured Regression**: *Given $A$ and $b$, with constant probability output a vector $x' \in \mathcal{C}$ for which*

$$\|T_q(A)x' - b\|_p \leq (1 + \varepsilon)\|T_q(A)x^\star - b\|_p,$$

*for an accuracy parameter $\varepsilon > 0$, where $x^\star = \arg\min_{x \in \mathcal{C}} \|T_q(A)x - b\|_p$.*

Our contributions in this paper are as follows:

○ For $p = 2$, we provide an algorithm that solves the structured regression problem above in time $O(\texttt{nnz}(A) \log^2 q) + \text{poly}(dq\epsilon^{-1})$. By combining our sketching methods with preconditioned iterative solvers, we can also obtain logarithmic dependence on $\epsilon$. For $p = 1$, we provide an algorithm with runtime $O(\texttt{nnz}(A) \log n \log^2 q) + \text{poly}(dq\epsilon^{-1} \log n)$. This implies that moving from linear (i.e, $Z = A$) to nonlinear regression ($Z = T_q(A)$)) incurs only a mild additional $\log^2 q$ runtime cost, while requiring no extra storage! Since $\texttt{nnz}(T_q(A)) = q \texttt{nnz}(A)$, this provides - to our knowledge - the first sketching approach that operates faster than "input-sparsity" time, i.e. we sketch $T_q(A)$ in time faster than $\texttt{nnz}(T_q(A))$.

○ Our algorithms apply to a broad class of nonlinear models for both least squares regression and their robust $l_1$ regression counterparts. While polynomial regression and additive models with monomial basis functions are immediately covered by our methods, we also show that under a suitable choice of the constraint set $\mathcal{C}$, the structured regression problem with $Z = T_q(AG)$ for a Gaussian random matrix $G$ approximates non-parametric regression using the Gaussian kernel. We argue that our approach provides a more flexible modeling framework when compared to randomized Fourier maps for kernel methods [16, 11].

○ Empirical results confirm both the practical value of our modeling framework, as well as speedup benefits of sketching.

## 2 Polynomial Fitting, Additive Models and Random Fourier Maps

Our primary goal in this section is to motivate sketching approaches for a versatile class of Block-Vandermonde structured regression problems by showing that these problems arise naturally in various statistical modeling settings.

The most basic application is the one-dimensional ($d = 1$) polynomial regression.

In multivariate additive regression models, a continuous target variable $y \in \mathbb{R}$ and input variables $z \in \mathbb{R}^d$ are related through the model $y = \mu + \sum_{i=1}^d f_i(z_i) + \epsilon_i$ where $\mu$ is an intercept term, $\epsilon_i$ are zero-mean Gaussian error terms and $f_i$ are smooth univariate functions. The basic idea is to expand each function as $f_i(\cdot) = \sum_{t=1}^q \beta_{i,t} h_{i,t}(\cdot)$ using basis functions $h_{i,t}(\cdot)$ and estimate the unknown parameter vector $x = [\beta_{11} \ldots \beta_{1q} \ldots \beta_{dq}]^T$ typically by a constrained or penalized least squares model, $\operatorname{argmin}_{x \in \mathcal{C}} \|Zx - b\|_2^2$ where $b = (y_1 \ldots y_n)^T$ and $Z = [H_1 \ldots H_q] \in \mathbb{R}^{n \times dq}$ for $(H_i)_{j,t} = h_{i,t}(z_j)$ on a training sample $(z_i, y_i), i = 1 \ldots n$. The constraint set $\mathcal{C}$ typically imposes smoothing, sparsity or group sparsity constraints [2]. It is easy to see that choosing a monomial basis $h_{i,s}(u) = u^s$ immediately maps the design matrix $Z$ to the structured regression form of Eqn. 2. For $p = 1$, our algorithms also provide fast solvers for robust polynomial additive models.

Additive models impose a restricted form of univariate nonlinearity which ignores interactions between covariates. Let us denote an interaction term as $z^\alpha = z_1^{\alpha_1} \ldots z_d^{\alpha_d}, \alpha = (\alpha_1 \ldots \alpha_d)$ where $\sum_i \alpha_i = q$, $\alpha_i \in \{0 \ldots q\}$. A degree-$q$ multivariate polynomial function space $P_q$ is spanned by $\{z^\alpha, \alpha \in \{0, \ldots q\}^d, \sum_i \alpha_i \leq q\}$. $P_q$ admits all possible degree-$q$ interactions but has dimensionality $d^q$ which is computationally infeasible to explicitly work with except for low-degrees and low-dimensional or sparse datasets [3]. Kernel methods with polynomial kernels $k(z, z') = (z^T z')^q = \sum_\alpha z^\alpha z'^\alpha$ provide an implicit mechanism to compute inner products in the feature space associated with $P_q$. However, they require $O(n^3)$ computation for solving associated kernelized (ridge) regression problems and $O(n^2)$ storage of dense $n \times n$ Gram matrices $K$ (given by $K_{ij} = k(z_i, z_j)$), and therefore do not scale well.

For a $d \times D$ matrix $G$ let $S_G$ be the subspace spanned by

$$\left\{ \left( \sum_{i=1}^d G_{ij} z_i \right)^t, t = 1 \ldots q, j = 1 \ldots s \right\} .$$

Assuming $D = d^q$ and that $G$ is a random matrix of i.i.d Gaussian variables, then almost surely we have $S_G = P_q$. An intuitively appealing explicit scalable approach is then to use $D \ll d^q$. In that case $S_G$ essentially spans a random subspace of $P_q$. The design matrix for solving the multivariate polynomial regression restricted to $S_G$ has the form $Z = T_q(AG)$ where $A = [z_1^T \ldots z_n^T]^T$.

This scheme can be in fact related to the idea of random Fourier features introduced by Rahimi and Recht [16] in the context of approximating shift-invariant kernel functions, with the Gaussian Kernel $k(z, z') = \exp\left(-\|z - z'\|_2^2 / 2\sigma^2\right)$ as the primary example. By appealing to Bochner's Theorem [18], it is shown that the Gaussian kernel is the Fourier transform of a zero-mean multivariate Gaussian distribution with covariance matrix $\sigma^{-1} I_d$ where $I_d$ denotes the $d$-dimensional identity matrix,

$$k(z, z') = \exp\left(-\|z - z'\|_2^2 / 2\sigma^2\right) = \mathbb{E}_{\omega \sim N(0_d, \sigma^{-1} I_d)}[\phi_\omega(z) \phi_\omega(z')^*]$$

where $\phi_\omega(z) = e^{i\omega' z}$. An empirical approximation to this expectation can be obtained by sampling $D$ frequencies $\omega \sim N(0_d, \sigma^{-1} I_d)$ and setting $k(z, z') = \frac{1}{D} \sum_{i=1}^D \phi_{\omega_i}(z) \phi_{\omega_i}(z)^*$. This implies that the Gram matrix of the Gaussian kernel, $K_{ij} = \exp\left(-\|z_i - z_j\|_2^2 / 2\sigma^2\right)$ may be approximated with high concentration as $K \approx RR^T$ where $R = [\cos(AG) \ \sin(AG)] \in \mathbb{R}^{n \times 2D}$ (sine and cosine are applied elementwise as scalar functions). This randomized explicit feature mapping for the Gaussian kernel implies that standard linear regression, with $R$ as the design matrix, can then be used to obtain a solution in time $O(nD^2)$. By taking the Maclaurin series expansion of sine and cosine upto degree $q$, we can see that a restricted structured regression problem of the form,

$\text{argmin}_{x \in \texttt{range}(Q)} \|T_q(AG)x - b\|_p$, where the matrix $Q \in \mathbb{R}^{2Dq \times 2D}$ contains appropriate coefficients of the Maclaurin series, will closely approximate the randomized Fourier features construction of [16]. By dropping or modifying the constraint set $x \in \texttt{range}(Q)$, the setup above, in principle, can define a richer class of models. A full error analysis of this approach is the subject of a separate paper.

## 3 Fast Structured Regression with Sketching

We now develop our randomized solvers for block-Vandermonde structured $l_p$ regression problems. In the theoretical developments below, we consider unconstrained regression though our results generalize straightforwardly to convex constraint sets $\mathcal{C}$. For simplicity, we state all our results for constant failure probability. One can always repeat the regression procedure $O(\log(1/\delta))$ times, each time with independent randomness, and choose the best solution found. This reduces the failure probability to $\delta$.

### 3.1 Background

We begin by giving some notation and then provide necessary technical background.

Given a matrix $M \in \mathbb{R}^{n \times d}$, let $M_1, \ldots, M_d$ be the columns of $M$, and $M^1, \ldots, M^n$ be the rows of $M$. Define $\|M\|_1$ to be the element-wise $\ell_1$ norm of $M$. That is, $\|M\|_1 = \sum_{i \in [d]} \|M_i\|_1$. Let $\|M\|_F = \left( \sum_{i \in [n], j \in [d]} M_{i,j}^2 \right)^{1/2}$ be the Frobenius norm of $M$. Let $[n] = \{1, \ldots, n\}$.

#### 3.1.1 Well-Conditioning and Sampling of A Matrix

**Definition 3 ($(\alpha, \beta, 1)$-well-conditioning [8])** *Given a matrix $M \in \mathbb{R}^{n \times d}$, we say $M$ is $(\alpha, \beta, 1)$-well-conditioned if (1) $\|x\|_\infty \leq \beta \|Mx\|_1$ for any $x \in \mathbb{R}^d$, and (2) $\|M\|_1 \leq \alpha$.*

**Lemma 4 (Implicit in [20])** *Suppose $S$ is an $r \times n$ matrix so that for all $x \in \mathbb{R}^d$,*

$$\|Mx\|_1 \leq \|SMx\|_1 \leq \kappa\|Mx\|_1.$$

*Let $Q \cdot R$ be a QR-decomposition of $SM$, so that $QR = SM$ and $Q$ has orthonormal columns. Then $MR^{-1}$ is $(d\sqrt{r}, \kappa, 1)$-well-conditioned.*

**Theorem 5 (Theorem 3.2 of [8])** *Suppose $U$ is an $(\alpha, \beta, 1)$-well-conditioned basis of an $n \times d$ matrix $A$. For each $i \in [n]$, let $p_i \geq \min\left(1, \frac{\|U_i\|_1}{t\|U\|_1}\right)$, where $t \geq 32\alpha\beta(d \ln\left(\frac{12}{\varepsilon}\right) + \ln\left(\frac{2}{\delta}\right))/(\varepsilon^2)$. Suppose we independently sample each row with probability $p_i$, and create a diagonal matrix $S$ where $S_{i,i} = 0$ if $i$ is not sampled, and $S_{i,i} = 1/p_i$ if $i$ is sampled. Then with probability at least $1 - \delta$, simultaneously for all $x \in \mathbb{R}^d$ we have:*

$$|\|SAx\|_1 - \|Ax\|_1| \leq \varepsilon\|Ax\|_1.$$

We also need the following method of quickly obtaining approximations to the $p_i$'s in Theorem 5, which was originally given in Mahoney et al. [13].

**Theorem 6** *Let $U \in \mathbb{R}^{n \times d}$ be an $(\alpha, \beta, 1)$-well-conditioned basis of an $n \times d$ matrix $A$. Suppose $G$ is a $d \times O(\log n)$ matrix of i.i.d. Gaussians. Let $p_i = \min\left(1, \frac{\|U_i G\|_1}{t2\sqrt{d}\|UG\|_1}\right)$ for all $i$, where $t$ is as in Theorem 5. Then with probability $1 - 1/n$, over the choice of $G$, the following occurs. If we sample each row with probability $p_i$, and create $S$ as in Theorem 5, then with probability at least $1 - \delta$, over our choice of sampled rows, simultaneously for all $x \in \mathbb{R}^d$ we have:*

$$|\|SAx\|_1 - \|Ax\|_1| \leq \varepsilon\|Ax\|_1.$$

#### 3.1.2 Oblivious Subspace Embeddings

Let $A \in \mathbb{R}^{n \times d}$. We assume that $n > d$. Let $\texttt{nnz}(A)$ denote the number of non-zero entries of $A$. We can assume $\texttt{nnz}(A) \geq n$ and that there are no all-zero rows or columns in $A$.

$\ell_2$ **Norm** The following family of matrices is due to Charikar et al. [4] (see also [9]): For a parameter $t$, define a random linear map $\Phi D : \mathbb{R}^n \to \mathbb{R}^t$ as follows:

- $h : [n] \mapsto [t]$ is a random map so that for each $i \in [n]$, $h(i) = t'$ for $t' \in [t]$ with probability $1/t$.
- $\Phi \in \{0,1\}^{t \times n}$ is a $t \times n$ binary matrix with $\Phi_{h(i),i} = 1$, and all remaining entries 0.
- $D$ is an $n \times n$ random diagonal matrix, with each diagonal entry independently chosen to be $+1$ or $-1$ with equal probability.

We will refer to $\Pi = \Phi D$ as a *sparse embedding matrix*.

For certain $t$, it was recently shown [7] that with probability at least .99 over the choice of $\Phi$ and $D$, for any fixed $A \in \mathbb{R}^{n \times d}$, we have simultaneously for all $x \in \mathbb{R}^d$,

$$(1 - \varepsilon) \cdot \|Ax\|_2 \le \|\Pi Ax\|_2 \le (1 + \varepsilon) \cdot \|Ax\|_2 \,,$$

that is, the entire column space of $A$ is preserved [7]. The best known value of $t$ is $t = O(d^2/\varepsilon^2)$ [14, 15] .

We will also use an oblivious subspace embedding known as the *subsampled randomized Hadamard transform*, or SRHT. See Boutsidis and Gittens's recent article for a state-the-art analysis [1].

**Theorem 7 (Lemma 6 in [1])** *There is a distribution over linear maps $\Pi'$ such that with probability .99 over the choice of $\Pi'$, for any fixed $A \in \mathbb{R}^{n \times d}$, we have simultaneously for all $x \in \mathbb{R}^d$,*

$$(1 - \varepsilon) \cdot \|Ax\|_2 \le \|\Pi' Ax\|_2 \le (1 + \varepsilon) \cdot \|Ax\|_2 \,,$$

*where the number of rows of $\Pi'$ is $t' = O(\varepsilon^{-2}(\log d)(\sqrt{d} + \sqrt{\log n})^2)$, and the time to compute $\Pi' A$ is $O(nd \log t')$.*

$\ell_1$ **Norm** The results can be generalized to subspace embeddings with respect to the $\ell_1$-norm [7, 14, 21]. The best known bounds are due to Woodruff and Zhang [21], so we use their family of embedding matrices in what follows. Here the goal is to design a distribution over matrices $\Psi$, so that with probability at least .99, for any fixed $A \in \mathbb{R}^{n \times d}$, simultaneously for all $x \in \mathbb{R}^d$,

$$\|Ax\|_1 \le \|\Psi Ax\|_1 \le \kappa \|Ax\|_1 \,,$$

where $\kappa > 1$ is a distortion parameter. The best known value of $\kappa$, independent of $n$, for which $\Psi A$ can be computed in $O(nnz(A))$ time is $\kappa = O(d^2 \log^2 d)$ [21]. Their family of matrices $\Psi$ is chosen to be of the form $\Pi \cdot E$, where $\Pi$ is as above with parameter $t = d^{1+\gamma}$ for arbitrarily small constant $\gamma > 0$, and $E$ is a diagonal matrix with $E_{i,i} = 1/u_i$, where $u_1, \ldots, u_n$ are independent standard exponentially distributed random variables.

Recall that an exponential distribution has support $x \in [0, \infty)$, probability density function (PDF) $f(x) = e^{-x}$ and cumulative distribution function (CDF) $F(x) = 1 - e^{-x}$. We say a random variable $X$ is exponential if $X$ is chosen from the exponential distribution.

### 3.1.3 Fast Vandermonde Multipication

**Lemma 8** *Let $x_0, \ldots, x_{n-1} \in \mathbb{R}$ and $V = V_{q,n}(x_0, \ldots, x_{n-1})$. For any $y \in \mathbb{R}^n$ and $z \in \mathbb{R}^q$, the matrix-vector products $Vy$ and $V^T z$ can be computed in $O((n + q) \log^2 q)$ time.*

### 3.2 Main Lemmas

We handle $\ell_2$ and $\ell_1$ separately. Our algorithms uses the subroutines given by the next lemmas.

**Lemma 9 (Efficient Multiplication of a Sparse Sketch and $T_q(A)$)** *Let $A \in \mathbb{R}^{n \times d}$. Let $\Pi = \Phi D$ be a sparse embedding matrix for the $\ell_2$ norm with associated hash function $h : [n] \to [t]$ for an arbitrary value of $t$, and let $E$ be any diagonal matrix. There is a deterministic algorithm to compute the product $\Phi \cdot D \cdot E \cdot T_q(A)$ in $O((nnz(A) + dtq) \log^2 q)$ time.*

**Proof:** By definition of $T_q(A)$, it suffices to prove this when $d = 1$. Indeed, if we can prove for a column vector $a$ that the product $\Phi \cdot D \cdot E \cdot T_q(a)$ can be computed in $O((nnz(a) + tq) \log^2 q)$ time, then by linearity if will follow that the product $\Phi \cdot D \cdot E \cdot T_q(A)$ can be computed in $O((nnz(A +$

---

**Algorithm 1** StructRegression-2

---

1: **Input:** An $n \times d$ matrix $A$ with $\mathtt{nnz}(A)$ non-zero entries, an $n \times 1$ vector $b$, an integer degree $q$, and an accuracy parameter $\varepsilon > 0$.
2: **Output:** With probability at least $.98$, a vector $x' \in \mathbb{R}^d$ for which $\|T_q(A)x' - b\|_2 \leq (1 + \varepsilon) \min_x \|T_q(A)x - b\|_2$.

3: Let $\Pi = \Phi D$ be a sparse embedding matrix for the $\ell_2$ norm with $t = O((dq)^2/\varepsilon^2)$.
4: Compute $\Pi T_q(A)$ using the efficient algorithm of Lemma 9 with $E$ set to the identity matrix.
5: Compute $\Pi b$.
6: Compute $\Pi'(\Pi T_q(A))$ and $\Pi' \Pi b$, where $\Pi'$ is a subsampled randomized Hadamard transform of Theorem 7 with $t' = O(\varepsilon^{-2}(\log(dq))(\sqrt{dq} + \sqrt{\log t})^2)$ rows.
7: Output the minimizer $x'$ of $\|\Pi' \Pi T_q(A)x' - \Pi' \Pi b\|_2$.

---

$dtq) \log^2 q)$ time for general $d$. Hence, in what follows, we assume that $d = 1$ and our matrix $A$ is a column vector $a$. Notice that if $a$ is just a column vector, then $T_q(A)$ is equal to $V_{q,n}(a_1, \ldots, a_n)^T$.

For each $k \in [t]$, define the ordered list $L^k = i$ such that $a_i \neq 0$ and $h(i) = k$. Let $\ell_k = |L^k|$. We define an $\ell_k$-dimensional vector $\sigma^k$ as follows. If $p_k(i)$ is the $i$-th element of $L^k$, we set $\sigma_i^k = D_{p_k(i),p_k(i)} \cdot E_{p_k(i),p_k(i)}$. Let $V^k$ be the submatrix of $V_{q,n}(a_1, \ldots, a_n)^T$ whose rows are in the set $L^k$. Notice that $V^k$ is itself the transpose of a Vandermonde matrix, where the number of rows of $V^k$ is $\ell_k$. By Lemma 8, the product $\sigma^k V^k$ can be computed in $O((\ell_k + q) \log^2 q)$ time. Notice that $\sigma^k V^k$ is equal to the $k$-th row of the product $\Phi D E T_q(a)$. Therefore, the entire product $\Phi D E T_q(a)$ can be computed in $O\left(\sum_k \ell_k \log^2 q\right) = O((\mathtt{nnz}(a) + tq) \log^2 q)$ time. ∎

**Lemma 10 (Efficient Multiplication of $T_q(A)$ on the Right)** *Let $A \in \mathbb{R}^{n \times d}$. For any vector $z$, there is a deterministic algorithm to compute the matrix vector product $T_q(A) \cdot z$ in $O((\mathtt{nnz}(A) + dq) \log^2 q)$ time.*

The proof is provided in the supplementary material.

**Lemma 11 (Efficient Multiplication of $T_q(A)$ on the Left)** *Let $A \in \mathbb{R}^{n \times d}$. For any vector $z$, there is a deterministic algorithm to compute the matrix vector product $z \cdot T_q(A)$ in $O((\mathtt{nnz}(A) + dq) \log^2 q)$ time.*

The proof is provided in the supplementary material.

### 3.3 Fast $\ell_2$-regression

We start by considering the structured regression problem in the case $p = 2$. We give an algorithm for this problem in Algorithm 1.

**Theorem 12** *Algorithm* STRUCTREGRESSION-2 *solves w.h.p the structured regression with $p = 2$ in time*

$$O(\mathtt{nnz}(A) \log^2 q) + \mathrm{poly}(dq/\varepsilon).$$

**Proof:** By the properties of a sparse embedding matrix (see Section 3.1.2), with probability at least $.99$, for $t = O((dq)^2/\varepsilon^2)$, we have simultaneously for all $y$ in the span of the columns of $T_q(A)$ adjoined with $b$,

$$(1 - \varepsilon)\|y\|_2 \leq \|\Pi y\|_2 \leq (1 + \varepsilon)\|y\|_2,$$

since the span of this space has dimension at most $dq + 1$. By Theorem 7, we further have that with probability $.99$, for all vectors $z$ in the span of the columns of $\Pi(T_q(A) \circ b)$,

$$(1 - \varepsilon)\|z\|_2 \leq \|\Pi' z\|_2 \leq (1 + \varepsilon)\|z\|_2.$$

It follows that for all vectors $x \in \mathbb{R}^d$,

$$(1 - O(\varepsilon))\|T_q(A)x - b\|_2 \leq \|\Pi' \Pi(T_q(A)x - B)\|_2 \leq (1 + O(\varepsilon))\|T_q(A)x - b\|_2.$$

It follows by a union bound that with probability at least $.98$, the output of STRUCTREGRESSION-2 is a $(1 + \varepsilon)$-approximation.

For the time complexity, $\Pi T_q(A)$ can be computed in $O((\mathtt{nnz}(A) + dtq) \log^2 q)$ by Lemma 9, while $\Pi b$ can be computed in $O(n)$ time. The remaining steps can be performed in $\mathrm{poly}(dq/\varepsilon)$ time, and therefore the overall time is $O(\mathtt{nnz}(A) \log^2 q) + \mathrm{poly}(dq/\varepsilon)$. ∎

---

**Algorithm 2** StructRegression-1

1: **Input:** An $n \times d$ matrix $A$ with $\mathtt{nnz}(A)$ non-zero entries, an $n \times 1$ vector $b$, an integer degree $q$, and an accuracy parameter $\varepsilon > 0$.
2: **Output:** With probability at least $.98$, a vector $x' \in \mathbb{R}^d$ for which $\|T_q(A)x' - b\|_1 \leq (1 + \varepsilon)\min_x \|T_q(A)x - b\|_1$.

3: Let $\Psi = \Pi E = \Phi D E$ be a subspace embedding matrix for the $\ell_1$ norm with $t = (dq + 1)^{1+\gamma}$ for an arbitrarily small constant $\gamma > 0$.
4: Compute $\Psi T_q(A) = \Pi E T_q(A)$ using the efficient algorithm of Lemma 9.
5: Compute $\Psi b = \Pi E b$.
6: Compute a QR-decomposition of $\Psi(T_q(A) \circ b)$, where $\circ$ denotes the adjoining of column vector $b$ to $T_q(A)$.
7: Let $G$ be a $(dq + 1) \times O(\log n)$ matrix of i.i.d. Gaussians.
8: Compute $R^{-1} \cdot G$.
9: Compute $(T_q(A) \circ b) \cdot (R^{-1}G)$ using the efficient algorithm of Lemma 10 applied to each of the columns of $R^{-1}G$.
10: Let $S$ be the diagonal matrix of Theorem 6 formed by sampling $\tilde{O}(q^{1+\gamma/2}d^{4+\gamma/2}\varepsilon^{-2})$ rows of $T_q(A)$ and corresponding entries of $b$ using the scheme of Theorem 6.
11: Output the minimizer $x'$ of $\|ST_q(A)x' - Sb\|_1$.

---

### 3.3.1 Logarithmic Dependence on $1/\varepsilon$

The STRUCTREGRESSION-2 algorithm can be modified to obtain a running time with a logarithmic dependence on $\varepsilon$ by combining sketching-based methods with iterative ones.

**Theorem 13** *There is an algorithm which solves the structured regression problem with $p = 2$ in time $O((\mathtt{nnz}(A) + dq)\log(1/\varepsilon)) + \mathrm{poly}(dq)$ w.h.p.*

Due to space limitations the proof is provided in Supplementary material.

### 3.4 Fast $\ell_1$-regression

We now consider the structured regression in the case $p = 1$. The algorithm in this case is more complicated than that for $p = 2$, and is given in Algorithm 2.

**Theorem 14** *Algorithm* STRUCTREGRESSION-1 *solves w.h.p the structured regression in problem with $p = 1$ in time*

$$O(\mathtt{nnz}(A)\log n \log^2 q) + \mathrm{poly}(dq\varepsilon^{-1}\log n).$$

The proof is provided in supplementary material.

We note when there is a convex constraint set $\mathcal{C}$ the only change in the above algorithms is to optimize over $x' \in \mathcal{C}$.

## 4 Experiments

We report two sets of experiments on classification and regression datasets. The first set of experiments compares generalization performance of our structured nonlinear least squares regression models against standard linear regression, and nonlinear regression with random fourier features [16]. The second set of experiments focus on scalability benefits of sketching. We used Regularized Least Squares Classification (RLSC) for classification.

Generalization performance is reported in Table 1. As expected, ordinary $\ell_2$ linear regression is very fast, especially if the matrix is sparse. However, it delivers only mediocre results. The results improve somewhat with additive polynomial regression. Additive polynomial regression maintains the sparsity structure so it can exploit fast sparse solvers. Once we introduce random features, thereby introducing interaction terms, results improve considerably. When compared with random Fourier features, for the same number of random features $D$, additive polynomial regression with random features get better results than regression with random Fourier features. If the number of random features is not the same, then if $D_{Fourier} = D_{Poly} \cdot q$ (where $D_{Fourier}$ is the number of Fourier features, and $D_{Poly}$ is the number of random features in the additive polynomial regression) then regression with random Fourier features seems to outperform additive polynomial regression with random features. However, computing the random features is one of the most expensive steps, so computing better approximations with fewer random features is desirable.

Figure 1 reports the benefit of sketching in terms of running times, and the trade-off in terms of accuracy. In this experiment we use a larger sample of the MNIST dataset with 300,000 examples.

| Dataset | Ord. Reg. | Add. Poly. Reg. | Add. Poly. Reg. w/ Random Features | Ord. Reg. w/ Fourier Features |
|---|---|---|---|---|
| MNIST classification $n = 60,000, d = 784$ $k = 10,000$ | 14% 3.9 sec | 11% 19.1 sec $q = 4$ | 6.9% 5.5 sec $D = 300, q = 4$ | 7.8% 6.8 sec $D = 500$ |
| CPU regression $n = 6,554, d = 21$ $k = 819$ | 12% 0.01 sec | 3.3% 0.07 sec $q = 4$ | 2.8% 0.13 sec $D = 60, q = 4$ | 2.8% 0.14 sec $D = 180$ |
| ADULT classification $n = 32,561, d = 123$ $k = 16,281$ | 15.5% 0.17 sec | 15.5% 0.55 sec $q = 4$ | 15.0% 3.9 sec $D = 500, q = 4$ | 15.1% 3.6 sec $D = 1000$ |
| CENSUS regression $n = 18,186, d = 119$ $k = 2,273$ | 7.1% 0.3 sec | 7.0% 1.4 sec $q = 4$ $\lambda = 0.2$ | 6.85% 1.9 sec $D = 500, q = 4,$ $\lambda = 0.1$ | 6.5% 2.1 sec $D = 500$ $\lambda = 0.1$ |
| FOREST COVER classification $n = 522,910, d = 54$ $k = 58,102$ | 25.7% 3.3 sec | 23.7% 7.8 sec $q = 4$ | 20.0% 14.0 sec $D = 200, q = 4$ | 21.3% 15.5 sec $D = 400$ |

Table 1: Comparison of testing error and training time of the different methods. In the table, $n$ is number of training instances, $d$ is the number of features per instance and $k$ is the number of instances in the test set. "Ord. Reg." stands for ordinary $\ell_2$ regression. "Add. Poly. Reg." stands for additive polynomial $\ell_2$ regression. For classification tasks, the percent of testing points incorrectly predicted is reported. For regression tasks, we report $\|y_p - y\|_2 / \|y\|$ where $y_p$ is the predicted values and $y$ is the ground truth.

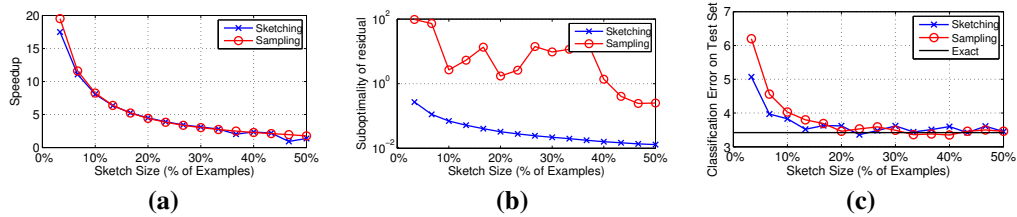

Figure 1: Examining the performance of sketching.

We compute 1,500 random features, and then solve the corresponding additive polynomial regression problem with $q = 4$, both exactly and with sketching to different number of rows. We also tested a sampling based approach which simply randomly samples a subset of the rows (no sketching). Figure 1 (a) plots the speedup of the sketched method relative to the exact solution. In these experiments we use a non-optimized straightforward implementation that does not exploit fast Vandermonde multiplication or parallel processing. Therefore, running times were measured using a sequential execution. We measured only the time required to solve the regression problem. For this experiment we use a machine with two quad-core Intel E5410 @ 2.33GHz, and 32GB DDR2 800MHz RAM. Figure 1 (b) explores the sub-optimality in solving the regression problem. More specifically, we plot $(\|Y_p - Y\|_F - \|Y_p^\star - Y\|_F)/\|Y_p^\star - Y\|_F$ where $Y$ is the labels matrix, $Y_p^\star$ is the best approximation (exact solution), and $Y_p$ is the sketched solution. We see that indeed the error decreases as the size of the sampled matrix grows, and that with a sketch size that is not too big we get to about a 10% larger objective. In Figure 1 (c) we see that this translates to an increase in error rate. Encouragingly, a sketch as small as 15% of the number of examples is enough to have a very small increase in error rate, while still solving the regression problem more than 5 times faster (the speedup is expected to grow for larger datasets).

## Acknowledgements

The authors acknowledge the support from XDATA program of the Defense Advanced Research Projects Agency (DARPA), administered through Air Force Research Laboratory contract FA8750-12-C-0323.

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
