[Supplementary Material]

# Supplementary Material: Sketching Structured Matrices For Faster Nonlinear Regression

**Haim Avron**
IBM T.J. Watson Research Center
Yorktown Heights, NY 10598
haimav@us.ibm.com

**Vikas Sindhwani**
IBM T.J. Watson Research Center
Yorktown Heights, NY 10598
vsindhw@us.ibm.com

**David P. Woodruff**
IBM Almaden Research Center
San Jose, CA 95120
dpwoodru@us.ibm.com

The supplementary material contains a complete version (including proofs) of Section 3 (Fast Structured Regression with Sketching).

## 1 Fast Structured Regression with Sketching

We now develop our randomized solvers for block-Vandermonde structured $l_p$ regression problems. In the theoretical developments below, we consider unconstrained regression though our results generalize straightforwardly to convex constraint sets $\mathcal{C}$.

### 1.1 Background

We begin by giving some notation and then provide necessary technical background.

Given a matrix $M \in \mathbb{R}^{n \times d}$, let $M_1, \ldots, M_d$ be the columns of $M$, and $M^1, \ldots, M^n$ be the rows of $M$. Define $\|M\|_1$ to be the element-wise $\ell_1$ norm of $M$. That is, $\|M\|_1 = \sum_{i \in [d]} \|M_i\|_1$. Let $\|M\|_F = \left( \sum_{i \in [n], j \in [d]} M_{i,j}^2 \right)^{1/2}$ be the Frobenius norm of $M$. Let $[n] = \{1, \ldots, n\}$.

#### 1.1.1 Well-Conditioning and Sampling of A Matrix

**Definition 1 ($(\alpha, \beta, 1)$-well-conditioning [7])** *Given a matrix $M \in \mathbb{R}^{n \times d}$, we say $M$ is $(\alpha, \beta, 1)$-well-conditioned if (1) $\|x\|_\infty \le \beta \|Mx\|_1$ for any $x \in \mathbb{R}^d$, and (2) $\|M\|_1 \le \alpha$.*

**Lemma 2 (Implicit in [14])** *Suppose $S$ is an $r \times n$ matrix so that for all $x \in \mathbb{R}^d$,*

$$\|Mx\|_1 \le \|SMx\|_1 \le \kappa \|Mx\|_1.$$

*Let $Q \cdot R$ be a QR-decomposition of $SM$, so that $QR = SM$ and $Q$ has orthonormal columns. Then $MR^{-1}$ is $(d\sqrt{r}, \kappa, 1)$-well-conditioned.*

**Proof:** For any standard unit vector $e_i$,

$$\|MR^{-1}e_i\|_1 \le \|SMR^{-1}e_1\|_1 \le \sqrt{r}\|SMR^{-1}e_i\|_2 = \sqrt{r},$$

and so $\|MR^{-1}\|_1 = \sum_i \|MR^{-1}e_i\|_1 \le d\sqrt{r}$. Also, for any $x$,

$$\kappa\|MR^{-1}x\|_1 \ge \|SMR^{-1}x\|_1 \ge \|SMR^{-1}x\|_2 = \|x\|_2 \ge \|x\|_\infty.$$

■

**Theorem 3 (Theorem 3.2 of [7])** *Suppose $U$ is an $(\alpha, \beta, 1)$-well-conditioned basis of an $n \times d$ matrix $A$. For each $i \in [n]$, let $p_i \geq \min\left(1, \frac{\|U_i\|_1}{t\|U\|_1}\right)$, where $t \geq 32\alpha\beta(d\ln\left(\frac{12}{\varepsilon}\right) + \ln\left(\frac{2}{\delta}\right))/(\varepsilon^2)$. Suppose we independently sample each row with probability $p_i$, and create a diagonal matrix $S$ where $S_{i,i} = 0$ if $i$ is not sampled, and $S_{i,i} = 1/p_i$ if $i$ is sampled. Then with probability at least $1 - \delta$, simultaneously for all $x \in \mathbb{R}^d$ we have:*

$$|\|SAx\|_1 - \|Ax\|_1| \leq \varepsilon\|Ax\|_1.$$

We also need the following method of quickly obtaining approximations to the $p_i$'s in Theorem 3, which was originally given in Mahoney et al. [10].

**Theorem 4** *Let $U \in \mathbb{R}^{n \times d}$ be an $(\alpha, \beta, 1)$-well-conditioned basis of an $n \times d$ matrix $A$. Suppose $G$ is a $d \times O(\log n)$ matrix of i.i.d. Gaussians. Let $p_i = \min\left(1, \frac{\|U_iG\|_1}{t2\sqrt{d}\|UG\|_1}\right)$ for all $i$, where $t$ is as in Theorem 3. Then with probability $1 - 1/n$, over the choice of $G$, the following occurs. If we sample each row with probability $p_i$, and create $S$ as in Theorem 3, then with probability at least $1 - \delta$, over our choice of sampled rows, simultaneously for all $x \in \mathbb{R}^d$ we have:*

$$|\|SAx\|_1 - \|Ax\|_1| \leq \varepsilon\|Ax\|_1.$$

**Proof:** Since $G$ is a $d \times O(\log n)$ matrix of i.i.d. Gaussians, we have that with probability at least $1 - 1/n$, over the choice of $G$, that

$$\|U_iG\|_1 \geq \|U_iG\|_2 \geq 1/2\|U_i\|_2 \geq 1/(2\sqrt{d})\|U_i\|_1,$$

simultaneously for all $i \in [n]$. The theorem now follows by Theorem 3. ∎

### 1.1.2 Oblivious Subspace Embeddings

Let $A \in \mathbb{R}^{n \times d}$. We assume that $n > d$. Let $\texttt{nnz}(A)$ denote the number of non-zero entries of $A$. We can assume $\texttt{nnz}(A) \geq n$ and that there are no all-zero rows or columns in $A$.

$\ell_2$ **Norm** The following family of matrices is due to Charikar et al. [5] (see also [8]): For a parameter $t$, define a random linear map $\Phi D : \mathbb{R}^n \to \mathbb{R}^t$ as follows:

- $h : [n] \mapsto [t]$ is a random map so that for each $i \in [n]$, $h(i) = t'$ for $t' \in [t]$ with probability $1/t$.
- $\Phi \in \{0,1\}^{t \times n}$ is a $t \times n$ binary matrix with $\Phi_{h(i),i} = 1$, and all remaining entries 0.
- $D$ is an $n \times n$ random diagonal matrix, with each diagonal entry independently chosen to be $+1$ or $-1$ with equal probability.

We will refer to $\Pi = \Phi D$ as a *sparse embedding matrix*.

For certain $t$, it was recently shown that with probability at least .99 over the choice of $\Phi$ and $D$, for any fixed $A \in \mathbb{R}^{n \times d}$, we have simultaneously for all $x \in \mathbb{R}^d$,

$$(1 - \varepsilon) \cdot \|Ax\|_2 \leq \|\Pi Ax\|_2 \leq (1 + \varepsilon) \cdot \|Ax\|_2,$$

that is, the entire column space of $A$ is preserved [6]. The best known value of $t$ is $t = O(d^2/\varepsilon^2)$ [11, 12] .

We will also use an oblivious subspace embedding known as the *subsampled randomized Hadamard transform*, or SRHT. See Boutsidis and Gittens's recent article for a state-the-art analysis [3].

**Theorem 5 (Lemma 6 in [3])** *There is a distribution over linear maps $\Pi'$ such that with probability .99 over the choice of $\Pi'$, for any fixed $A \in \mathbb{R}^{n \times d}$, we have simultaneously for all $x \in \mathbb{R}^d$,*

$$(1 - \varepsilon) \cdot \|Ax\|_2 \leq \|\Pi'Ax\|_2 \leq (1 + \varepsilon) \cdot \|Ax\|_2,$$

*where the number of rows of $\Pi'$ is $t' = O(\varepsilon^{-2}(\log d)(\sqrt{d} + \sqrt{\log n})^2)$, and the time to compute $\Pi'A$ is $O(nd\log t')$.*

We note that there are implicit connections between these embedding results and the compressed sensing literature, and the well-known restricted isometry property (RIP); see, for example, [4, 1]

$\ell_1$ **Norm**   The results can be generalized to subspace embeddings with respect to the $\ell_1$-norm [6, 11, 16]. The best known bounds are due to Woodruff and Zhang [16], so we use their family of embedding matrices in what follows. Here the goal is to design a distribution over matrices $\Psi$, so that with probability at least .99, for any fixed $A \in \mathbb{R}^{n \times d}$, simultaneously for all $x \in \mathbb{R}^d$,

$$\|Ax\|_1 \leq \|\Psi Ax\|_1 \leq \kappa \|Ax\|_1 \,,$$

where $\kappa > 1$ is a distortion parameter. The best known value of $\kappa$, independent of $d$, is $\kappa = O(d^2 \log^2 d)$ [16]. Their family of matrices $\Psi$ is chosen to be of the form $\Pi \cdot E$, where $\Pi$ is as above with parameter $t = d^{1+\gamma}$ for arbitrarily small constant $\gamma > 0$, and $E$ is a diagonal matrix with $E_{i,i} = 1/u_i$, where $u_1, \ldots, u_n$ are independent standard exponentially distributed random variables.

Recall that an exponential distribution has support $x \in [0, \infty)$, probability density function (PDF) $f(x) = e^{-x}$ and cumulative distribution function (CDF) $F(x) = 1 - e^{-x}$. We say a random variable $X$ is exponential if $X$ is chosen from the exponential distribution.

Again, we note the implicit connection to the notion of $\ell_1$-RIP [2].

### 1.1.3   Fast Vandermonde Multipication

**Lemma 6**  *Let $x_0, \ldots, x_{n-1} \in \mathbb{R}$ and $V = V_{q,n}(x_0, \ldots, x_{n-1})$. For any $y \in \mathbb{R}^n$ and $z \in \mathbb{R}^q$, the matrix-vector products $Vy$ and $V^T z$ can be computed in $O((n + q) \log^2 q)$ time.*

**Proof:**   It is known that if $n \leq q$, then $Vy$ and $V^T z$ can be computed in $O(q \log^2 q)$ time [13, 15, Theorem 2.11]. For $n > q$, write $n = \alpha q + \beta$, where $\alpha$ is a non-negative integer and $0 \leq \beta < q$. The lemma follows in general by writing

$$\left[ V = \ V_{q,q}(x_0, \ldots, x_{q-1}) \mid V_{q,q}(x_q, \ldots, x_{2q-1}) \mid \cdots \mid V_{q,j}(x_{iq}, \ldots, x_n) \ \right],$$

and computing $Vy$ or $V^T z$ using block multiplication. ∎

### 1.2   Main Lemmas

We handle $p = 2$ and $p = 1$ separately below. Our algorithms make use of the following subroutines given by our next lemmas.

**Lemma 7 (Efficient Multiplication of a Sparse Sketch and $T_q(A)$)**  *Let $A \in \mathbb{R}^{n \times d}$. Let $\Pi = \Phi D$ be a sparse embedding matrix for the $\ell_2$ norm with associated hash function $h : [n] \to [t]$ for an arbitrary value of $t$, and let $E$ be any diagonal matrix. There is a deterministic algorithm to compute the product $\Phi \cdot D \cdot E \cdot T_q(A)$ in $O((\text{nnz}(A) + dtq) \log^2 q)$ time.*

**Proof:**   By definition of $T_q(A)$, it suffices to prove this when $d = 1$. Indeed, if we can prove for a column vector $a$ that the product $\Phi \cdot D \cdot E \cdot T_q(a)$ can be computed in $O((\text{nnz}(a) + tq) \log^2 q)$ time, then by linearity if will follow that the product $\Phi \cdot D \cdot E \cdot T_q(A)$ can be computed in $O((\text{nnz}(A + dtq) \log^2 q)$ time for general $d$. Hence, in what follows, we assume that $d = 1$ and our matrix $A$ is a column vector $a$. Notice that if $a$ is just a column vector, then $T_q(A)$ is equal to $V_{q,n}(a_1, \ldots, a_n)^T$.

For each $k \in [t]$, define the ordered list $L^k = (i$ such that $a_i \neq 0$ and $h(i) = k)$. Let $\ell_k = |L^k|$. We define an $\ell_k$-dimensional vector $\sigma^k$ as follows. If $p_k(i)$ is the $i$-th element of $L^k$, we set

$$\sigma_i^k = D_{p_k(i), p_k(i)} \cdot E_{p_k(i), p_k(i)}.$$

Let $V^k$ be the submatrix of $V_{q,n}(a_1, \ldots, a_n)^T$ whose rows are in the set $L^k$. Notice that $V^k$ is itself the transpose of a Vandermonde matrix, where the number of rows of $V^k$ is $\ell_k$. By Theorem 6, the product $\sigma^k V^k$ can be computed in $O((\ell_k + q) \log^2 q)$ time. Notice that $\sigma^k V^k$ is equal to the $k$-th row of the product $\Phi DET_q(a)$. Therefore, the entire product $\Phi DET_q(a)$ can be computed in

$$O \left( \sum_k \ell_k \log^2 q \right) = O((\text{nnz}(a) + tq) \log^2 q)$$

time. ∎

---
**Algorithm 1** StructRegression-2
---
1: **Input:** An $n \times d$ matrix $A$ with $\mathtt{nnz}(A)$ non-zero entries, an $n \times 1$ vector $b$, an integer degree $q$, and an accuracy parameter $\varepsilon > 0$.
2: **Output:** With probability at least $.98$, a vector $x' \in \mathbb{R}^d$ for which $\|T_q(A)x' - b\|_2 \leq (1+\varepsilon)\min_x \|T_q(A)x - b\|_2$.

3: Let $\Pi = \Phi D$ be a sparse embedding matrix for the $\ell_2$ norm with $t = O((dq)^2/\varepsilon^2)$.
4: Compute $\Pi T_q(A)$ using the efficient algorithm of Lemma 7 with $E$ set to the identity matrix.
5: Compute $\Pi b$.
6: Compute $\Pi'(\Pi T_q(A))$ and $\Pi'\Pi b$, where $\Pi'$ is a subsampled randomized Hadamard transform of Theorem 5 with $t' = O(\varepsilon^{-2}(\log(dq))(\sqrt{dq} + \sqrt{\log t})^2)$ rows.
7: Output the minimizer $x'$ of $\|\Pi'\Pi T_q(A)x' - \Pi'\Pi b\|_2$.
---

**Lemma 8 (Efficient Multiplication of $T_q(A)$ on the Right)** *Let $A \in \mathbb{R}^{n \times d}$. For any vector $z$, there is a deterministic algorithm to compute the matrix vector product $T_q(A) \cdot z$ in $O((\mathtt{nnz}(A) + dq)\log^2 q)$ time.*

**Proof:** We will prove that for a column vector $a$, that $T_q(a) \cdot z^i$, where $z^i$ denotes the $i$-th block of $q$ coordinates of $z$, can be computed in $O((\mathtt{nnz}(a) + q)\log^2 q)$ time. Moreover, $T_q(a) \cdot z^i$ will be an $n$-dimensional vector with $\mathtt{nnz}(a)$ non-zero entries. This will be sufficient to establish the lemma since we have $T_q(A) \cdot z = \sum_{i=1}^d T_q(A_i) \cdot z^i$. Then if $T_q(A_i)z^i$ can be computed in $O((\mathtt{nnz}(A_i) + dq)\log^2 q)$ time, it follows that in $O((\mathtt{nnz}(A) + dq)\log^2 q)$ time, we can compute $T_q(A_i)z^i$ simultaneously for all $i$. Moreover, as each $T_q(A_i)z^i$ is $n$-dimensional and has $O(\mathtt{nnz}(A_i))$ non-zero entries, the resulting vectors can be added together in $O(\mathtt{nnz}(A))$ time.

It remains to prove the claimed result for a column vector $a = (a_1, \ldots, a_n)$. Notice that $T_q(a)$ is equal to $V_{q,n}(a_1, \ldots, a_n)^T$. Let $L = (i \text{ such that } a_i \neq 0 \text{ and })$, and $\ell = |L| = \mathtt{nnz}(a)$. Let $V$ be the submatrix of $V_{q,n}(a_1, \ldots, a_n)^T$ containing the rows in $L$. Note that $V$ is itself the transpose of a Vandermonde matrix, where the number of rows of $V$ is $\ell$. Let $z_L^i$ be the $\ell$-dimensional vector whose $j$-th entry equals the $j$-th non-zero coordinate in $z^i$. By Theorem 6, the product $Vz_L^i$ can be computed in $O((\ell + q)\log^2 q)$ time. Notice that the non-zero entries of $T_q(a) \cdot z^i$ are exactly the entries of $Vz_L^i$, where the $j$-th entry of $T_q(a) \cdot z$ equals the entry of $Vz_L$ corresponding to the $j$-th entry in $L$. It follows that we have computed $T_q(a) \cdot z^i$ in $O((\mathtt{nnz}(a) + q)\log^2 q)$ time and is an $n$-dimensional vector with $\mathtt{nnz}(a)$ non-zero entries. ∎

**Lemma 9 (Efficient Multiplication of $T_q(A)$ on the Left)** *Let $A \in \mathbb{R}^{n \times d}$. For any vector $z$, there is a deterministic algorithm to compute the matrix vector product $z \cdot T_q(A)$ in $O((\mathtt{nnz}(A) + dq)\log^2 q)$ time.*

**Proof:** We will prove that for a column vector $a$, that $zT_q(a)$ can be computed in $O((\mathtt{nnz}(a) + q)\log^2 q)$ time. This will be sufficient to establish the lemma since then the overall time complexity is $\sum_{i=1}^d O((\mathtt{nnz}(A_i) + q)\log^2 q) = O((\mathtt{nnz}(A) + dq)\log^2 q)$.

It remains to prove the claimed result for a column vector $a = (a_1, \ldots, a_n)$. Notice that $T_q(a)$ is equal to $V_{q,n}(a_1, \ldots, a_n)^T$. Let $L = (i \text{ such that } a_i \neq 0 \text{ and })$, and $\ell = |L| = \mathtt{nnz}(a)$. Let $V$ be the submatrix of $V_{q,n}(a_1, \ldots, a_n)^T$ containing the rows in $L$. Note that $V$ is itself the transpose of a Vandermonde matrix, where the number of rows of $V$ is $\ell$. Let $z_L$ be the $\ell$-dimensional vector whose $j$-th entry equals the $j$-th non-zero coordinate in $z$. By Theorem 6, the product $z_L \cdot V$ can be computed in $O((\ell + q)\log^2 q) = O((\mathtt{nnz}(a) + q)\log^2 q)$ time. ∎

## 1.3 Fast $\ell_2$-regression

We start by considering the structured regression problem in the case $p = 2$. We give an algorithm for this problem in Figure **??**.

**Theorem 10** *Algorithm* STRUCTREGRESSION-2 *solves w.h.p the structured regression with $p = 2$ in time*

$$O(\mathtt{nnz}(A)\log^2 q) + \mathrm{poly}(dq/\varepsilon).$$

**Proof:** By the properties of a sparse embedding matrix (see Section 1.1.2), with probability at least .99, for $t = O((dq)^2/\varepsilon^2)$, we have simultaneously for all $y$ in the span of the columns of $T_q(A)$ adjoined with $b$,

$$(1 - \varepsilon)\|y\|_2 \le \|\Pi y\|_2 \le (1 + \varepsilon)\|y\|_2,$$

since the span of this space has dimension at most $dq + 1$. By Theorem 5, we further have that with probability .99, for all vectors $z$ in the span of the columns of $\Pi(T_q(A) \circ b)$,

$$(1 - \varepsilon)\|z\|_2 \le \|\Pi' z\|_2 \le (1 + \varepsilon)\|z\|_2.$$

It follows that for all vectors $x \in \mathbb{R}^d$,

$$(1 - O(\varepsilon))\|T_q(A)x - b\|_2 \le \|\Pi'\Pi(T_q(A)x - B)\|_2 \le (1 + O(\varepsilon))\|T_q(A)x - b\|_2.$$

It follows by a union bound that with probability at least .98, the output of STRUCTREGRESSION-2 is a $(1 + \varepsilon)$-approximation.

For the time complexity, $\Pi T_q(A)$ can be computed in $O((\text{nnz}(A) + dtq)\log^2 q)$ by Lemma 7, while $\Pi b$ can be computed in $O(n)$ time. The remaining steps can be performed in $\text{poly}(dq/\varepsilon)$ time, and therefore the overall time is $O(\text{nnz}(A)\log^2 q) + \text{poly}(dq/\varepsilon)$. ∎

### 1.3.1 Logarithmic Dependence on $1/\varepsilon$

The STRUCTREGRESSION-2 algorithm can be modified to obtain a running time with a logarithmic dependence on $\varepsilon$ by combining sketching-based methods with iterative ones.

The analysis follows that of Section 7.7 of [6], but here we additionally use the fast right matrix-vector multiplication algorithm associated with Vandermonde matrices in the iterative algorithm. Define the condition number $\kappa(B^\top B) = \frac{\sup_{x, \|x\| = 1}\|Bx\|^2}{\inf_{x, \|x\| = 1}\|Bx\|^2}$, and let $x^0, x^1, \ldots$ be the estimates generated by CG on $B^\top B$ with righthand side equal to $B^\top b$. It is well-known that

$$\frac{\left\|B(x^{(m)} - x^\star)\right\|^2}{\left\|B(x^{(0)} - x^\star)\right\|^2} \le 2\left(\frac{\sqrt{\kappa(B^\top B)} - 1}{\sqrt{\kappa(B^\top B)} + 1}\right)^m. \tag{1}$$

where $B^\top B x^\star = B^\top b$ [9, Theorem 10.2.6]. Thus the running time depends on the condition number. The running time per iteration is the time needed to compute matrix-vector products $Bx$ and $B^\top x$, plus $O(n + d)$ for vector arithmetic. Here we set $B = T_q(A)$ for an input matrix $A$. By Lemma 8, for a vector $x$, given $A$ and $x$ the matrix-vector product $T_q(A) \cdot z$ can be computed in $O((\text{nnz}(A) + dq)\log^2 q)$ time.

Suppose we run STRUCTREGRESSION-2 with constant $\varepsilon = \varepsilon_0$. Let $Q \cdot R$ be a QR-decomposition of $\Pi'\Pi T_q(A)$. Then since $\|\Pi'\Pi T_q(A)x\|_2 = (1 \pm \varepsilon_0)\|T_q(A)x\|_2$ for all $x \in \mathbb{R}^d$, that is $\Pi'\Pi$ is a subspace embedding for $\ell_2$, we have for any unit $x \in \mathbb{R}^d$,

$$\|T_q(A) \cdot R^{-1} x\|_2 \le \frac{1}{1 - \varepsilon_0}\|\Pi'\Pi T_q(A)R^{-1}x\|_2 = \frac{1}{1 - \varepsilon_0},$$

where the equality uses that $\Pi'\Pi T_q(A) = Q \cdot R$ where $Q$ has orthonormal columns. Similarly,

$$\|T_q(A) \cdot R^{-1} x\|_2 \ge \frac{1}{1 + \varepsilon_0}\|\Pi'\Pi T_q(A)R^{-1}x\|_2 = \frac{1}{1 + \varepsilon_0}.$$

It follows that the condition number

$$\kappa(T_q(A)R^{-1}) \le \frac{(1 + \varepsilon_0)^2}{(1 - \varepsilon_0)^2}.$$

That is, $T_q(A)R^{-1}$ is well-conditioned. Plugging this into (1), after $m$ iterations $\left\|AR(x^{(m)} - x^*)\right\|^2$ is at most $2\varepsilon_0^m$ times its starting value. Starting with a solution $x^{(0)}$ with relative error at most 1, and applying $1 + \log(1/\varepsilon)$ iterations of a conjugate-gradient like method with $\varepsilon_0 = 1/e$, the relative error is reduced to $\varepsilon$ and the total work is $O((\text{nnz}(A) + dq)\log(1/\varepsilon)) + \text{poly}(dq)$. We summarize this derivation in the following theorem.

**Theorem 11** *There is an algorithm which solves the structured regression problem with $p = 2$ in time $O((\text{nnz}(A) + dq)\log(1/\varepsilon)) + \text{poly}(dq)$ w.h.p.*

---
**Algorithm 2** StructRegression-1
---
1: **Input:** An $n \times d$ matrix $A$ with $\text{nnz}(A)$ non-zero entries, an $n \times 1$ vector $b$, an integer degree $q$, and an accuracy parameter $\varepsilon > 0$.
2: **Output:** With probability at least .98, a vector $x' \in \mathbb{R}^d$ for which $\|T_q(A)x' - b\|_1 \leq (1 + \varepsilon) \min_x \|T_q(A)x - b\|_1$.

3: Let $\Psi = \Pi E = \Phi D E$ be a subspace embedding matrix for the $\ell_1$ norm with $t = (dq + 1)^{1+\gamma}$ for an arbitrarily small constant $\gamma > 0$.
4: Compute $\Psi T_q(A) = \Pi E T_q(A)$ using the efficient algorithm of Lemma 7.
5: Compute $\Psi b = \Pi E b$.
6: Compute a QR-decomposition of $\Psi(T_q(A) \circ b)$, where $\circ$ denotes the adjoining of column vector $b$ to $T_q(A)$.
7: Let $G$ be a $(dq + 1) \times O(\log n)$ matrix of i.i.d. Gaussians.
8: Compute $R^{-1} \cdot G$.
9: Compute $(T_q(A) \circ b) \cdot (R^{-1}G)$ using the efficient algorithm of Lemma 8 applied to each of the columns of $R^{-1}G$.
10: Let $S$ be the diagonal matrix of Theorem 4 formed by sampling $\tilde{O}(q^{1+\gamma/2}d^{4+\gamma/2}\varepsilon^{-2})$ rows of $T_q(A)$ and corresponding entries of $b$ using the scheme of Theorem 4.
11: Output the minimizer $x'$ of $\|ST_q(A)x' - Sb\|_1$.
---

## 1.4 Fast $\ell_1$-regression

We now consider the structured regression in the case $p = 1$. The algorithm in this case is more complicated than that for $p = 2$, and is given in Figure (**??**).

**Theorem 12** *Algorithm* STRUCTREGRESSION-1 *solves w.h.p the structured regression in problem with $p = 1$ in time*

$$O(\text{nnz}(A) \log n \log^2 q) + \text{poly}(dq\varepsilon^{-1}\log n).$$

**Proof:** By the properties of a subspace embedding matrix for $\ell_1$ (see Section 1.1.2), with probability at least .99, for $t = (dq + 1)^{1+\gamma}$, we have simultaneously for all $y$ in the span of the columns of $T_q(A)$ adjoined with $b$,

$$\|y\|_1 \leq \|\Psi y\|_1 \leq \kappa\|y\|_1,$$

where $\kappa = O(d^2 \log^2 d)$. By Theorem 4, we further have that with probability .99, for all vectors in the span of the columns of $T_q(A)$ adjoined with $b$,

$$(1 - \varepsilon)\|y\|_1 \leq \|Sy\|_1 \leq (1 + \varepsilon)\|y\|_1.$$

It follows by a union bound that with probability at least .98, the output of GENADDITIVE-1 is a $(1 + \varepsilon)$-approximation.

For the time complexity, $\Psi T_q(A)$ can be computed in $O((\text{nnz}(A) + tdq) \log^2 q)$ time by Lemma 7, while $\Psi b$ can be computed in $O(n)$ time. Steps 4-6 can be performed in $\text{poly}(dq \log n)$ time. By Lemma 8, Step 7 can be performed in $O((\text{nnz}(A) + dq) \log^2 q \log n)$ time. Step 8 can be computed in $O(n \log n)$ time, and Step 9 can be done in $\text{poly}(dq\varepsilon^{-1})$ time. ∎

**Remark (Constrained Regression)**: Here we note the simple, though useful, observation that our algorithms also solve the constrained version of regression in which there is a constraint set $\mathcal{C}$ and we require $x \in \mathcal{C}$. Indeed, the only change is in Step 5 of STRUCTREGRESSION-2 and Step 9 of STRUCTREGRESSION-1 to instead compute the minimizer $x'$ over $x' \in \mathcal{C}$. Since $|\Pi'\Pi T_q(A)x' - \Pi'\Pi b|_2 = (1 \pm \varepsilon)\|T_q(A)x' - b\|_2$ for all $x$ in STRUCTREGRESSION-2, and since $\|ST_q(A)x' - Sb\|_1 = (1 \pm \varepsilon)\|T_q(A)x' - b\|_1$ for all $x$ in STRUCTREGRESSION-1, this is valid.