[Reviews · NeurIPS 2013]

Submitted by Assigned_Reviewer_5

The authors develop fast algorithms to solve the \ell_1 and \ell_2 linear regression problems, when the design matrix is block-Vandermonde. The authors show that matrix multiplication with a block-Vandermonde matrix can be very efficient (Lemmas 9, 10, and 11), which, with the help of the random embedding matrices proposed in [1] and [11], allows fast algorithms for solving the particular regression problems under consideration. Applications include polynomial fitting, additive models, and random Fourier maps.

Quality
Basically this paper is well-structured, but there is some unnecessary redundancy.

For example, only the first of the two problems for motivating this work seems relevant to the reviewer, while the second one looks quite artificial. A lot of existing results mentioned in Section 3 can be removed without sacrificing the completeness of this paper, and some proofs appearing in the main text are repeated again in the supplementary material.

Clarity
The results are stated in an imprecise manner.

For example, in Theorems 7 and 12, 13, and 14, the authors should characterize the convergence rate of the probabilities with respect to the matrix dimensions precisely, instead of merely saying that the probability is 0.99. And in Theorems 12, 13, and 14, the authors claim the proposed algorithms solve the regression problems under consideration, but do not mention that the solution are not exact, and with a small probability the solution errors may be unacceptable.

Originality
The main theoretical framework is not new, but basically based on the random embedding matrices proposed in [1] and [11]. The only novel contribution of this paper is to show that multiplication with a block-Vandermonde matrix is fast, and thus solving regression problems with the help of the embedding matrices proposed in [1] and [11], can be fast. This contribution is incremental in the reviewer’s opinion.

Significance
As stated, the theoretical contribution of this work is limited because of the originality issue.
Summary: The authors show that matrix multiplication with a block-Vandermonde matrix can be very efficient (Lemmas 9, 10, and 11), which, with the help of the random embedding matrices proposed in [1] and [11], allows fast algorithms for solving the particular regression problems under consideration.

Submitted by Assigned_Reviewer_6

This paper proposes efficient randomization-based algorithms for solving two types of regression problems -- corresponding to linear least squares and least absolute deviation fitting over convex sets -- in settings where the design matrix has a Vandermonde structure. To the reviewer's best knowledge, incorporating this type of structure into such randomized algorithmic approaches to linear algebra problems is novel. Such methods are timely and relevant, given many recent applications focused on processing and inference in "big data" regimes.

The paper is overall well-written and could, in the opinion of the reviewer, comprise a solid contribution to the NIPS community, subject to some minor revisions, described in some detail here.

Primary concerns:

1) The authors appear unaware of some connections between the subspace embedding they describe (claimed introduced in [6]) and existing efforts in the sketching literature. Specifically, the matrix structure described in section 3.1.2 here was previously proposed as a counting sketch in (Charikar, Moses, Kevin Chen, and Martin Farach-Colton. "Finding frequent items in data streams." Automata, Languages and Programming. Springer Berlin Heidelberg, 2002. 693-703.); see also, e.g., the recent survey article (Gilbert, Anna, and Piotr Indyk. "Sparse recovery using sparse matrices." Proceedings of the IEEE 98.6 (2010): 937-947.).

2) Further, there are implicit connections between the embedding results on page 5 and the compressive sensing literature. Namely, ensuring that the l_2 norm of vectors Ax remain preserved is related under action by a linear operator (here, $\Pi$) is related to the well-known restricted isometry property, albeit in this case "restricted" to a single subspace of dimension equal to the number of columns in A; see, for example, (Candès, Emmanuel J. "The restricted isometry property and its implications for compressed sensing." Comptes Rendus Mathematique 346.9 (2008): 589-592.), as well as (Baraniuk, Richard, et al. "A simple proof of the restricted isometry property for random matrices." Constructive Approximation 28.3 (2008): 253-263.) Likewise, the l_1 preservation relation on page 5 here is related to the l_1-RIP condition studied in (Berinde, Radu, et al. "Combining geometry and combinatorics: A unified approach to sparse signal recovery." Communication, Control, and Computing, 2008 46th Annual Allerton Conference on. IEEE, 2008.).

3) I found the experimental section somewhat unclear. Specifically, it was not obvious to me specifically what the classification approach was here. It would be helpful if the authors could explain this in some more detail.

Minor points:

1) In several places, including Theorem 6 and Algorithm 2, the authors describe a matrix of Gaussians. Is there any reason to prefer, e.g., unit variance here?

2) On page 5, the authors state "The best known value of $\kappa$, independent of $d$, is $\kappa = O(d^2 \log^2d)." How is this independent of d?

3) On page 6, "Let L = (i such that a_i \neq 0 and ), and ..." typo?

4) Table 1 in the "CENSUS" data set, the time is given as 1.4% sec

One Final General Suggestion

It may be helpful for the reader to better appreciate the contributions here, if the authors listed in each step of the proposed algorithm the corresponding computational complexity. This would be especially helpful where, e.g., the procedure requires QR decompositions -- this step gave the reviewer pause, at first viewing.


Summary: The paper is overall well-written and could, in the opinion of the reviewer, comprise a solid contribution to the NIPS community, subject to the revisions detailed above.


Submitted by Assigned_Reviewer_8

NA
Summary: NA
Author Feedback

Author rebuttal: REVIEWER 1 (Assigned_Reviewer_5):

We can trim down section 3, without sacrificing completeness, and remove proofs that are repeated in the supplementary material. We will also state the convergence rate, that is the dependence on the error probability delta, though in related papers in the literature delta is often taken to be a fixed constant, since one can repeat the regression procedure with independent randomness log 1/delta times to decrease the error probability to delta.

We are not sure we understand why references [1] and [11] are stated as the main ones which enable this speedup for structured regression. Here we critically use the subspace embedding initially proposed in [6] for achieving input sparsity time embeddings. We observe that this subspace embedding not only has the property used in [6], but also has the property that it can be applied to a block Vandermonde matrix very quickly, in time faster than would be necessary if using the analysis of [6] as is. That is, the subspace embedding of [6] has additional important properties, useful for polynomial fitting among other things, which as far as we are aware do not hold for other subspace embeddings, e.g., those used in [1] and [11].

REVIEWER 2 (Assigned_Reviewer_6):
- Thanks for pointing out the references - we should and will cite these works in the sketching, streaming, and compressed sensing literature. (NOTE: The references on compressed sensing were added only to the supplementary material due to lack of space).

- Regarding the experiments: we set up a regression problem for each class, setting in the right-hand side +1 for members of the class, and -1 for member of other classes. We then solve these c (number of classes) regression problem to obtain w_1, ..., w_c. Now, for a test vector x we first apply any possible non-linear transformation phi that was applied to the training set and take the inner product with each of the w_i's. That is compute phi(x)^T w_1, ..., phi(x)^T w_c. We then select for this vector the class i that maximized phi(x)^T w_i. This approach to classification is also referred to as Regularized Least Squares Classifier (RLSC) in the literature. We will be happy to clarify this point in the paper.

Regarding the minor points:
1) Unit variance Gaussians are what is intended. That is, the Gaussians are used to estimate the norm of a vector v, and if g is a vector of unit Gaussians, then < g, v > is a Gaussian with variance |v|_2^2, so we can estimate |v|_2 using this. If we use Gaussians with other variance sigma^2, then < g,v > would have variance sigma^2 |v|_2^2, so we could also estimate |v|_2 from this quantity, by looking at the expectation of < g,v > ^2 and dividing by sigma^2.

2) Thanks, that is a typo - should say independent of n.

3) Thanks, that is a typo - should just say "Let L = (i such that a_i \neq 0),"

4) Thanks, there should not be a % sign there.

We will list out the time complexities of the various steps. As for the QR decomposition, the main thing here is that it is done in a space of dimension that does not depend on n. But yes, we should spell it out.

REVIEWER 3:

Please note that we do report relative error in objective values with respect to the exact solution, as a function of sketch size in Figure 1b. We do wish to point out in Figure 1c that sketching is empirically very effective also for a downstream application metric, an analysis that is often missing in sketching literature, and one that would be of interest to NIPS practitioners. Computational performance is reported in Figure 1a, and a comparison of four methods on five datasets is provided in table 1. Our empirical study can definitely be made more exhaustive. We believe its inclusion will lead to greater practical interest in the paper. However, if the reviewer feels strongly enough about deferring the experimental section to future work, we will definitely give it due consideration.

Regarding the comments towards making improvements:

One: We will be more clear on what we meant by faster than input sparsity, and didn't mean to cause confusion. In fact, we can remove the claim. The overall algorithm is not faster than input sparsity time, which would be impossible to guarantee relative error, provided the input is represented in a concise way. We just meant, as in our response to Reviewer 1, that an interesting previously unexplored property of the mapping of [6] is that it can be multiplied with a fixed matrix A in time faster than nnz(A) provided that A has additional structure, which arise in realistic settings. We think such properties are worth further exploration.

Two: We think the reviewer is referring to the log^2 q here, and why it doesn't make things worse. This is because this is multiplying an nnz(A) term, but the matrix we are applying our sketch to is T_q(A), so previous results would have a larger nnz(T_q(A)) term. We can definitely clarify this.

Three: Regarding the constrained regression, we meant that to approximately solve min_x |Ax-b|_p subject to x in C, one can instead solve min_x |SAx-Sb|_p subject to x in C, where S is a sketching matrix. In the case of l_2, S is just a subspace embedding. In the case of l_1, S is a sampling and rescaling matrix which is formed by sampling rows according to their norm in a well-conditioned basis. Note that since for all x, |SAx-Sb|_p = (1+eps)|Ax-b|_p, this in particular holds for all x in C.